# Incidence and predictors of loss to follow up among adult HIV patients on antiretroviral therapy in University of Gondar Comprehensive Specialized Hospital: A competing risk regression modeling

**Achamyeleh Birhanu Teshale*, Adino Tesfahun Tsegaye, Haileab Fekadu Wolde**

Department of Epidemiology and Biostatistics, Institute of Public Health, College of Medicine and Health Sciences, University of Gondar, Gondar, Ethiopia

* achambir08@gmail.com

## Abstract

### Introduction

Loss to follow up after the initiation of antiretroviral therapy (ART) is common in Africa, particularly in Ethiopia and it is a considerable obstacle for the effectiveness of the ART program. Mortality is a competing risk of loss to follow up but it is often overlooked and there is limited evidence about the incidence and predictors of loss to follow up in the presence of competing events.

### Objective

To assess the Incidence and predictors of loss to follow up among adult HIV patients on ART in University of Gondar Comprehensive Specialized Hospital between January 1, 2015, and December 31, 2018.

### Methods

Institution based retrospective follow up study was conducted in University of Gondar Comprehensive Specialized Hospital. A Gray's test and cumulative incidence curve were used to compare the cumulative incidence function of loss to follow up. Bivariable and multivariable competing risk regression models were fitted to identify the predictors of lost to follow up and those variables with p-value <0.05 in the multivariable analysis was considered as significant predictors of lost to follow up.

### Result

A total of 531 adult HIV patients on ART were included in the analysis. The incidence rate of loss to follow up in this study was 10.90 (95% CI: 8.9–13.2) per 100 person years. Being age group 15–30 years (aSHR = 2.01; 95%CI;1.11–3.63), being daily laborer(aSHR = 2.60; 95% CI;1.45–4.66), not receiving cotrimoxazole preventive therapy (aSHR = 2.66; 95%CI;1.68–

**Data Availability Statement:** All relevant data are within the manuscript and its Supporting Information files.

**Funding:** The author received the fund or financial support from University of Gondar.

**Competing interests:** The authors have declared that no competing interests exist.

**Abbreviations:** ABC- 3TC-EF, Abacavir-Lamivudine-Efavirenz; ART, Antiretroviral Therapy; aSHR, Adjusted Sub distribution Hazard Ratio; AZT-3TC-EFV, Zidovudine-Lamivudine-Efavirenz; AZT-3TC- NVP, Zidovudine -Lamivudine -Nevirapine; CIF, Cumulative Incidence Function; CPT, Cotrimoxazole Preventive Therapy; cSHR, Crude Sub distribution Hazard Ratio; HIV, Human Immune Deficiency Virus; IPT, Isoniazid Preventive Therapy; LTFU, Lost To Follow Up; TDF-3TC-EVF, Tenofovir- Lamivudine-Efavirenz; WHO, World Health Organization.

4.21), not receiving isoniazid preventive therapy(aSHR = 4.57; 95% CI;1.60–13.08), ambulatory functional status (aSHR = 1.61; 95% CI; 1.02–2.51) and taking AZT-3TC-NVP medication at start of ART(aSHR = 2.01; 95% CI; 1.16–3.78) were significant predictors of lost to follow up.

## Conclusion

In this study the incidence of lost to follow up was high. Young people, daily laborer, ambulatory patients and those taking AZT-3TC-NVP as well as those who did not take opportunistic prophylaxis were at higher risk of loss to follow up. Therefore, giving special attention to the high-risk groups for lost to follow up highlighted in this study could decrease the rate of LTFU.

## Introduction

Access to antiretroviral therapy has increased rapidly since 2005 and globally an estimated 21.7 million people are receiving antiretroviral therapy (ART) in which the World Health Organization (WHO) African Region accounts 60% [1]. In East and Southern Africa, the average adult ART coverage is 66% [2]. In Ethiopia, the overall ART coverage is 54% of which the adult ART coverage accounts 58% [3].

Antiretroviral therapy has significantly reduced mortality and improved the life expectancy of HIV infected patients but the success still critically depends on regular patient follow up [4–6]. The loss to follow-up (LTFU) among HIV infected patients is related to ART adherence and is becoming an increasing problem in sub-Saharan Africa as the ART program is expanding; this has resulted in a decrease in the clinician-to-patient ratio [7,8]. It also accounts for the majority of all attrition and the problem of attrition can be addressed if one identifies the contributing factors and effectively tracks patients [6,9].

According to a study done in one of the countries in Sub Saharan Africa, Uganda the proportion of LTFU is 24.6% [10]. According to different retrospective follow up studies done in Ethiopia the incidence rate of LTFU ranges from 8.2 to 11.6 per 100 person-years [11–13]. The proportion of LTFU is also different across different regions of Ethiopia. According to many studies in Ethiopia, the proportion of LTFU ranges from 11.5 to 26.7% [11,13–16].

Loss to follow up of clients from ART have a great negative impact on the immunological benefits of ART, increase acquired immune deficiency syndrome (AIDS) related morbidity, mortality and hospitalization and it also results in serious consequences such as discontinuation of treatment, drug toxicity, treatment failure due to poor adherence and drug resistance [17–20]. Thus, High rates of LTFU from treatment programs pose a serious challenge to program implementers and constitutes an inefficient use of scarce treatment resources [21].

Different studies showed that LTFU is associated with base line sociodemographic factors like sex [10,16,22–24], age [10,11,13,25,26], educational status [26,27], marital status [25,28,29], occupation [30,31], disclosure status [16,27], distance from the health facility [29], caregiver [32,33] and clinical and treatment related factors like baseline WHO stage [25,34], baseline CD4 count [13,31,35], history of opportunistic infection at enrollment [10], baseline functional status [15,32,35–37], opportunistic prophylaxis [11,13,14,29] and type of ART regimen at start of their medication [23,31].

Even though there are many studies done on LTFU and its predictors, valid estimates of incidence and predictors of loss to follow-up can be obtained if one considers death as a competing event (rather than counting those who died as censored) especially in poor clinical settings in which death is common and alters the probability of the occurrence of loss to follow up. However, in most of these studies death which is a competing risk of loss to follow up is often overlooked and this may produce misleading results. Therefore, this study aimed to estimate the incidence rate and to identify the predictors of LTFU by considering death as a competing event in University of Gondar Comprehensive Specialized Hospital, Northwest, Ethiopia.

## Method

### Study design and setting

An institution-based retrospective follow-up study was conducted in University of Gondar Comprehensive Specialized Hospital between January 1, 2015, and December 31, 2018. The Hospital is found in Gondar town which is located 727 km from the capital city of the country, Addis Ababa and 172 km far from Bahir Dar, the capital city of the Amhara regional state. It is a leading referral hospital in Northwest Ethiopia serving more than five million people. ART service is one of the services given by this hospital and a summary of medical records of the hospital shows that currently, there are 5,573 patients on ART follow up among these 5,273 are adults.

### Sample size determination and sampling method

To check the sufficiency of samples, a minimum sample size (531) was determined using the power cox command of Stata 14 software. Record of study participants has been filtered first from the ART database according to their entry time to the follow-up, age and inclusion criteria. Finally, we select our study sample using a simple random sampling technique by R software.

### Variable definitions and data collection procedures

The study population was all HIV-infected adults (age≥15 years) who enrolled at University of Gondar Comprehensive Specialized Hospital ART clinic and who had at least one follow-up visit between January 1, 2015, and December 31, 2018. Those patients who had unknown ART initiation date and transferred in with incomplete baseline data were excluded from the study. The primary outcome variable was loss to follow up (LTFU) defined as not taking ART refill for 3 months or longer from the last attendance for refill and not yet classified as dead or transferred-out [29]. The competing event was death which was defined as the death of a patient. A patient was classified as censored if he/she had a formally recorded transfer to another clinic or still on follow up at this hospital at the end of the study period. The predictor variables assessed were baseline socio-demographic factors (sex, age, marital status, educational status, occupation, residence, distance from health facility, disclosure status and caregiver) and baseline clinical and treatment-related factors (past opportunistic infection, baseline CD4 count, baseline functional status, type of regimen at start, isoniazid and cotrimoxazole preventive therapy, baseline and last known WHO clinical stage, viral load, BMI and current TB status). Here the functional was defined based on the ART guideline; Working: able to perform usual work inside or outside the home, Ambulatory: able to perform an activity of daily living, and bedridden: not able to perform an activity of daily living [38]. Disclosure in this study was defined as disclosure of the status that is being HIV positive to at least one individual. In

addition, in this study caregiver was also defined as anyone who can support or assist the individual with HIV. The data were collected from the patient charts by one health officer and three clinical nurses by using a data extraction sheet which was designed based on study objectives. To control the data quality, training was given for the data collectors and the supervisor about the ways of extracting the data based on the study objectives. The tool was also pretested and the data were checked for consistency and completeness on a daily basis by the supervisor and principal investigator.

## Data processing and analysis

The Data was entered using Epi-data version 3.1 and exported to Stata 14 and R 3.5.3 software for analysis. Descriptive statistics including proportions, median, tables, and charts was done to describe the characteristics of the study participants. Nonparametric estimation of cumulative incidence function (CIF) was done both graphically and using Gray's test. After fitting the model, the proportional sub distribution hazard assumption was also checked by using the plot of log (- log (1-CIF)) versus the log of time to failure for each covariate, by interacting each covariate with time and using Schoenfeld residual test. Bivariable competing regression analysis was fitted to identify factors associated with LTFU. Those variables with a p-value of <0.2 in the bivariable analysis were again fitted to the multivariable competing risk regression analysis. Both crude and adjusted sub distribution hazard ratio with the corresponding 95% Confidence Interval (CI) was calculated to show the strength of association. In multivariable analysis, variables with a P-value of <0.05 were considered statistically significant.

## Ethical consideration

Ethical approval was obtained from the Institutional Review Board of University of Gondar. Since this study used an analysis of secondary data from patient charts, we received a waiver for informed consent. To keep the confidentiality names and other personal identifiers were not included in the data collection tool.

## Result

### Baseline socio-demographic characteristics

A total of 531 HIV infected adults on ART were included in the final analysis. Of these, 303 (57.4%), of participants were females. The median age of the study participants was 32 (IQR = 25–40) years. Nearly half 262 (49.34%) of the study participants had secondary and above educational status. Most 445(85.7%) of the study participants disclosed their HIV status to at least one person and 502(94.5%) of subjects had a caregiver (Table 1).

### Clinical and treatment-related characteristics

Among the study subjects, 221 (41.62%) had baseline CD4 count $\leq$200 cell/mm3 with median baseline CD4 count of 250cell/mm3 (IQR = 110 to 429 cell/mm3). Based on the baseline WHO staging 238 (44.82%) of patients were stage I followed by stage III patients,119(22.41%). Most 374 (70.4%) of the participants had a working functional status and 473 (89.08%) of study participants were taking TDF-3TC-EFV at the start of their medication. Regarding prophylaxis against opportunistic infections, 236(87.45%) of the respondents were on co-trimoxazole Preventive Therapy (CPT) and 123 (23.16%) were on isoniazid preventive therapy (IPT) (Table 2).

**Table 1. Baseline socio-demographic characteristics for adult HIV positive patients on ART in University of Gondar Comprehensive Specialized Hospital between January 1, 2015, and December 31, 2018.**

| Variables | Category | Frequency | Percentage (%) |
|---|---|---|---|
| Sex | Female | 303 | 57.44 |
| | Male | 226 | 42.56 |
| Age(years) | 15–30 | 249 | 46.89 |
| | 31–39 | 135 | 25.42 |
| | ≥40 | 147 | 27.86 |
| Marital status | Single | 114 | 21.47 |
| | Married | 245 | 46.14 |
| | Divorced | 132 | 24.86 |
| | Widowed | 40 | 7.53 |
| Occupation | Employee | 248 | 46.7 |
| | Daily labor | 75 | 14.12 |
| | Other* | 208 | 39.17 |
| Religion | Orthodox | 462 | 87.01 |
| | Muslim | 57 | 10.73 |
| | Other** | 12 | 2.26 |
| Caregiver | Yes | 502 | 94.54 |
| | No | 29 | 5.46 |
| Disclosure status | Disclosed | 445 | 83.80 |
| | Not disclosed | 86 | 16.20 |
| Educational status | No formal education | 128 | 24.11 |
| | Primary education | 141 | 26.55 |
| | Secondary and above | 262 | 49.34 |

Other* = driver, house wife, jobless and student, other** = protestant and catholic

## Incidence of LTFU and death

Study subjects were followed for a median of 19.6 months (IQR: 8.4–34) after initiation of treatment with a total observation time of 935.75 person-years.

During the four years follow up period, 51(9.6%) were dead and 102(19.21%) had been LTFU (Fig 1). The overall incidence of death and LTFU were 5.4 (95% CI: 4.1–7.2), and 10.90 (95% CI: 8.9,13.2) per 100-person year respectively. Lost to follow up was highest in the first 12 months of ART follow up,15.1per100 person-year(95%CI:11.9,19.2).

## Predictors of LTFU among adult HIV patients on ART

**Non-parametric estimation of cumulative incidence function of LTFU.** Non parametrically, CIFs across groups were checked statistically using Gray's test (which is analogous to the log-rank test of normal survival analysis) and graphically by plotting each predictor variable against failure time.

Based on the result of the modified $X^2$ test (Gray's test), there was a significant difference in CIF among categories of age, marital status, occupation, education, IPT, CPT, Functional status, disclosure status, baseline WHO stage, caregiver and type of regiment at the start of ART.

Graphically, belonging to the age group 15–30, being single, having an occupation other than that of an employee, not having caregiver, not taking IPT and CPT prophylaxis, being ambulatory, not having disclosed their HIV status, a baseline WHO stage IV, being on a

**Table 2. Clinical and treatment-related characteristics of adult HIV patients on ART at University of Gondar Comprehensive Specialized Hospital between January 1, 2015, and December 31, 2018.**

| Variable | Category | Frequency | Percentage (%) |
|---|---|---|---|
| Baseline CD4 count | ≤200 | 221 | 41.62 |
| | 201–350 | 117 | 22.03 |
| | >350 | 193 | 36.35 |
| Baseline WHO stage | Stage I | 238 | 44.82 |
| | Stage II | 101 | 19.02 |
| | Stage III | 119 | 22.41 |
| | Stage IV | 73 | 13.75 |
| Last known WHO stage | Stage I | 301 | 56.69 |
| | Stage II | 136 | 25.61 |
| | Stage III | 58 | 10.92 |
| | Stage IV | 36 | 6.78 |
| Past OIs | Yes | 225 | 42.37 |
| | No | 306 | 57.63 |
| CPT | Yes | 366 | 68.93 |
| | No | 165 | 31.07 |
| IPT | Yes | 123 | 23.16 |
| | No | 408 | 76.84 |
| Baseline Functional status | Working | 374 | 70.43 |
| | Ambulatory | 121 | 22.79 |
| | Bedridden | 36 | 6.78 |
| Type of regimen at start | 1e (TDF-3TC-EFV) | 473 | 89.08 |
| | 1c (AZT-3TC-NVP) | 39 | 7.34 |
| | 1d/1f/1g | 19 | 3.58 |
| Baseline BMI | <18.5 | 170 | 32.02 |
| | 18.5–24.9 | 314 | 59.13 |
| | ≥25 | 47 | 8.85 |
| Current TB status | Positive | 30 | 5.65 |
| | Negative | 501 | 94.35 |

1d = AZT-3TC-EFV,1f = TDF-3TC-NVP,1g = ABC- 3TC-EFV, BMI = body mass index, OI = opportunistic infection, CPT = cotrimoxazole preventive therapy, IPT = isoniazid preventive therapy

regimen of AZT-3TC-NVP and not having a formal education were all risk factors for LTFU (Fig 2).

**The multivariable competing risk regression model.** After fitting a bivariable competing risk regression model all the predictor variables except sex and baseline CD4 count were found to have p-value <0.2 and entered into multivariable analysis and variables such as age, occupation, regimen type at start, ambulatory functional status, IPT and CPT were found to be significant predictors for lost to follow up at 5% level of significance.

In our study keeping other variables constant, the sub hazard of LTFU is 2.01 times higher among adults whose age is between 15–30 years as compared to those whose age is ≥40 years (aSHR = 2.01; 95% CI: 1.11,3.63). Looking at occupation, the sub hazard of LTFU is 2.60 times higher among adults whose occupation is daily labor compared to those whose occupation is an employee (aSHR = 2.60; 95% CI: 1.45,4.66). Regarding CPT prophylaxis, the sub hazard of LTFU is 2.66 times higher among adults who are not taking CPT compared to their counterparts (aSHR = 2.66; 95% CI:1.68,4.21). The sub hazard of LTFU is 4.57 times higher among

## Survival status

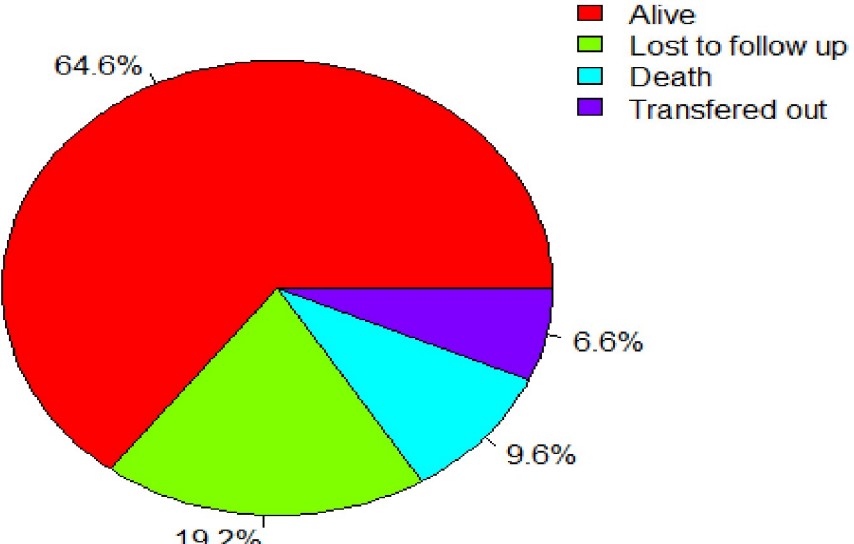

**Fig 1. Proportion of survival status among adult HIV patients on ART in University of Gondar Comprehensive Specialized Hospital between January 1, 2015, and December 31, 2018.**

adults who are not taking IPT compared to their counterparts (aSHR = 4.57; 95% CI: 1.60,13.08). Keeping other variables constant, the sub hazard of LTFU is 2.1 times higher among adults who are taking regimen AZT-3TC-NVP compared to those taking TDF-3TC-EFV (aSHR = 2.10; 95% CI: 1.16,3.78). Moreover, the sub hazard of LTFU is 1.61 times among adults who have an ambulatory functional status at enrollment as compared to those who have a working functional status (aSHR = 1.61;95%CI:1.02,2.51) (Table 3).

## Discussion

This study mainly assessed the incidence and predictors of LTFU among adult HIV patients on ART in University of Gondar Comprehensive Specialized Hospital, northwest Ethiopia. Many other studies reported different predictors for LTFU and our study also assessed socio-demographic, clinical and treatment-related factors. In our study factors such as age, occupation, history of taking IPT, CPT, baseline functional status and regimen type at ART initiation were found to be significantly associated with LTFU.

The incidence estimated in this study, which is10.90 per 100 person year, was consistent with studies done in different areas of Ethiopia [11,13], South Africa [30] and Cameroon [29]. This might be due to the implementation of ART services according to the WHO ART guideline. However, it was lower than a study done in Asian-pacific [39]. This is because this study counts LTFU once for a single participant but there may be double or triple counting of patients who are LTFU since there may be reentering and re-LTFU and contribute to more than one episode of LTFU which increases the incidence (in case of Asian Pacific study). But it was higher than other studies done in Mekelle-Ethiopia [23], India [28] and Asia [40]. Lost to follow up in our study turned out to be higher than the study done in Mekelle might be due to the study period in which currently, starting from 2016, test and treat strategy is introduced in Ethiopia and this might increase the number of patients on follow up and which in turn

## Graphical estimation of cumulative incidence functions for selected variables

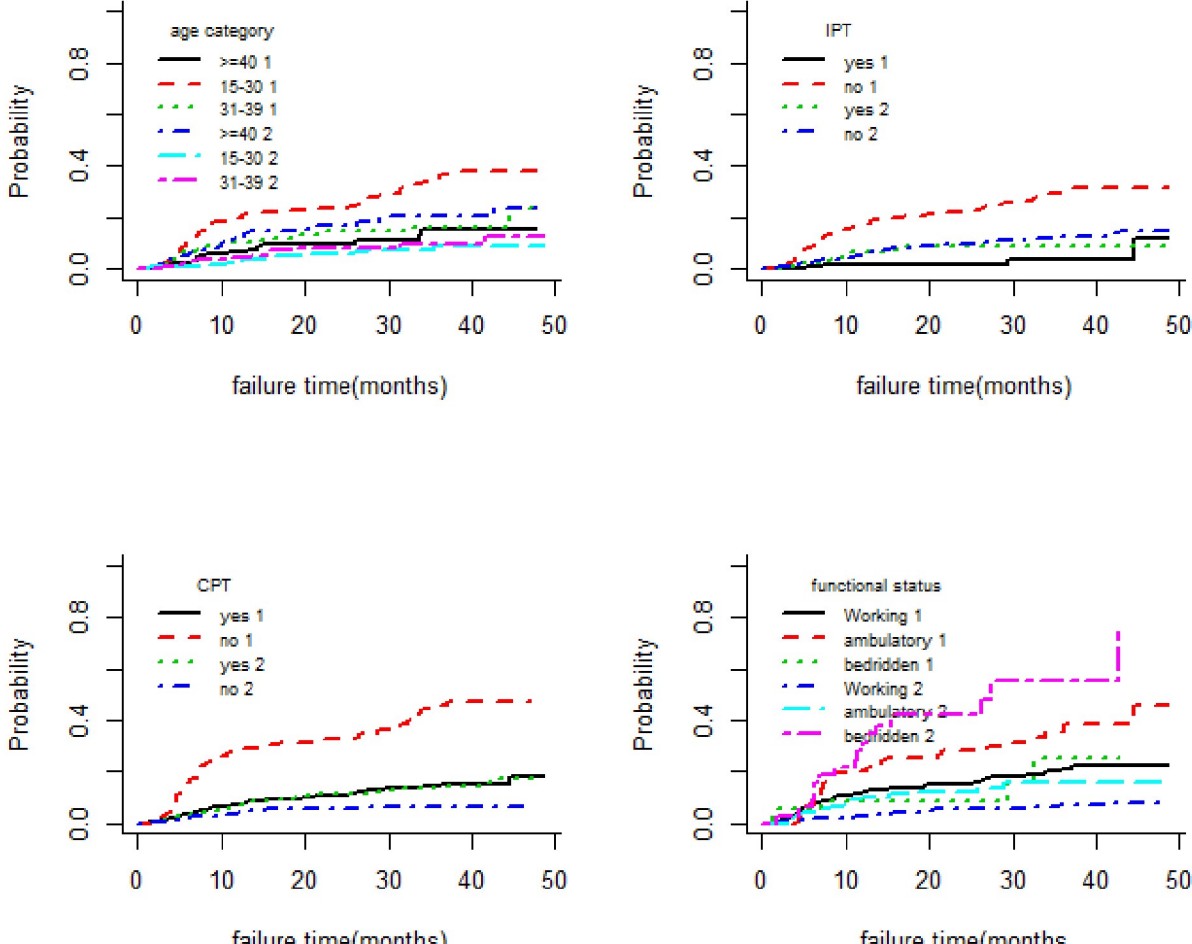

**Fig 2. Non-parametric estimates of cumulative incidence curves with LTFU (1) and death (2) as a competing event for selected variable categories of age, IPT, functional status, and disclosure.**

increases self-referral to the other health facilities, as investigated by different studies [24,41]. In addition, in search of literatures, patient satisfaction with healthcare system is associated with engagement in HIV care [32,42], so there may be the difficulty of appropriate health service delivery and access since this hospital is a Comprehensive Specialized Hospital, which serves many patients and finally results in patient dissatisfaction and loss to follow up. Furthermore, the way operationalizing LTFU might be a reason for higher LTFU in our study as compared to studies in India and Asia because in our study LTFU is defined when a patient is lost for at least 90 days but in studies in India and Asia LTFU is defined when the patient is lost for at least 180 days.

The younger age groups were more likely to be LTFU in ART as compared to older patients. This result is congruent with different studies done in different regions of Ethiopia [11,13], Nigeria [25], Sub Saharan Africa [10] and Guinea [26]. This finding might be because

**Table 3. Bivariable and multivariable competing risk regression analysis for predictors of LTFU among adult HIV patients on ART at University of Gondar Comprehensive Specialized Hospital between January 1, 2015, and December 31, 2018.**

| Variable | Category | Survival status | | | cSHR [95% CI] | aSHR [95% CI] |
| --- | --- | --- | --- | --- | --- | --- |
| | | Censored (378) | LTFU (102) | Death (51) | | |
| Age (years) | 15–30 | 167 | 67 | 15 | 2.85(1.63–4.98) | 2.01(1.11–3.63) * |
| | 31–39 | 103 | 20 | 12 | 1.35(0.69–2.65) | 1.06(0.51–2.21) |
| | ≥40 | 108 | 15 | 24 | 1 | 1 |
| Marital status | Single | 67 | 38 | 9 | 2.67(1.68–4.21) | 1.51(0.90–2.53) |
| | Married | 192 | 35 | 18 | 1 | 1 |
| | Divorced | 92 | 24 | 16 | 1.34(0.80–2.26) | 1.14(0.65–2.02) |
| | Widowed | 27 | 5 | 51 | 0.79(0.31–2.03) | 0.77(0.30–1.97) |
| Occupation | Employee | 194 | 30 | 24 | 1 | 1 |
| | Daily labor | 44 | 26 | 5 | 3.39(2.00–5.74) | 2.60(1.45–4.66) * |
| | Other | 140 | 46 | 22 | 1.98(1.25–3.13) | 1.38(0.81–2.33) |
| Educational level | No education | 83 | 35 | 10 | 1.74(1.12–2.69) | 1.27(0.78–2.08) |
| | Primary education | 104 | 20 | 17 | 0.78(0.46–1.32) | 0.62(0.35–1.08) |
| | Sec.& above | 191 | 47 | 24 | 1 | 1 |
| Caregiver | Yes | 362 | 92 | 48 | 1 | 1 |
| | No | 16 | 10 | 3 | 2.39(1.22–4.69) | 1.05(0.45–2.46) |
| Disclosure | Disclosed | 334 | 71 | 40 | 1 | 1 |
| | Not disclosed | 44 | 31 | 11 | 2.58(1.71–3.91) | 1.56(0.92–2.58) |
| Past OIs | Yes | 240 | 51 | 34 | 1 | 1 |
| | No | 238 | 51 | 17 | 0.72(0.49–1.07) | 0.87(0.54–1.42) |
| CPT | Yes | 282 | 42 | 42 | 1 | 1 |
| | No | 96 | 63 | 9 | 3.68(2.49–5.45) | 2.66(1.68–4.21)** |
| IPT | Yes | 110 | 4 | 9 | 1 | 1 |
| | No | 268 | 98 | 42 | 7.81(2.9–21.0) | 4.57(1.60–13.08) ** |
| Functional status | Working | 292 | 62 | 20 | 1 | 1 |
| | Ambulatory | 71 | 35 | 15 | 1.91(1.27–2.87) | 1.61(1.02–2.51)* |
| | Bedridden | 15 | 5 | 16 | 0.87(0.35–2.17) | 0.43(0.17–1.12) |
| Baseline WHO stage | Stage I/II | 264 | 57 | 18 | 1 | 1 |
| | Stage III | 77 | 27 | 15 | 1.34(0.86–2.1) | 1.17(0.67–2.04) |
| | Stage IV | 37 | 18 | 18 | 1.64(0.95–2.83) | 1.58(0.78–3.20) |
| Last WHO stage | Stage I/II | 362 | 79 | 16 | 1 | 1 |
| | Stage III/IV | 36 | 23 | 35 | 1.47(0.91–2.36) | 1.42(0.79–2.55) |
| Current TB status | Yes | 12 | 9 | 9 | 1 | 1 |
| | No | 366 | 93 | 42 | 0.64(0.32–1.25) | 1.36(0.56–3.29) |
| BMI | <18.5 | 111 | 41 | 18 | 1.53(1.01–2.30) | 1.14(0.71–1.83) |
| | 18.5–24.9 | 231 | 53 | 30 | 1 | 1 |
| | ≥25 | 36 | 8 | 3 | 0.94(0.45–1.97) | 0.85(0.35–2.08) |
| Regimen type at the start | 1e (TDF-3TC EFV) | 348 | 82 | 43 | 1 | 1 |
| | 1c(AZT-3TC-NVP) | 20 | 15 | 4 | 2.08(1.21–3.58) | 2.10(1.16–3.78) * |
| | 1d/1f/1g | 10 | 5 | 4 | 1.26(0.52–3.07) | 0.75(0.34–1.66) |

cSHR = crude sub hazard ratio, aSHR = adjusted sub hazard ratio,

*pvalue<0.05,

**pvalue<0.01,

1d = AZT-3TC-EFV,1f = TDF-3TC-NVP,1g = ABC- 3TC-EFV, BMI = body mass index

this group of population is either dependent on others or they are more mobile as compared to the older population and also in this study most of this group of the population had no care-giver which closely monitor their follow-up which finally may increase their tendency to LTFU. In addition, in this group of population there is fear of stigma and discrimination and due to this they preferred to enroll in care facilities located relatively far from their neighbor-hoods, they may be far away from this institution, and there may be additional costs in time and transportation which affects their regular follow up [31].

In our study patients whose occupation is daily labor increases the risk of LTFU and this is supported by a study done in the Oromia region of Ethiopia [31]. This might be since daily laborers pass the whole day in the workplace and do not have enough time to come to the health facilities for follow up. Because of the nature of their work, they are also more mobile and have no constant workplace that may end up with lost from follow up [43]. Besides, this group of population had low income (unable to feed themselves) and they believe that taking ART medication on an empty stomach is dangerous and finally this leads them to seek alterna-tive healing options such as holy water by stopping ART and becomes lost to follow up [32,44].

In the current study, those patients who were not taking IPT prophylaxis were at higher risk of LTFU. This finding is supported by different studies done in Ethiopia [11,13,14]. This might be due to, initiation of IPT which is recommended by the national ART guideline might directly or indirectly decrease LTFU by increasing retention of HIV patients on care since IPT prevents the occurrence of tuberculosis co-infection which is the most frequent life-threaten-ing opportunistic disease among people living with HIV [38]. In addition, this might be because the most commonly mentioned factor motivating people to stay in care is improved health [32] and this might be happening through appropriate interventions such as manage-ment of opportunistic infections. Moreover, in this study patients who were not taking CPT were more likely to be LTFU from ART and this is in line with a study done in Cameroon [29]. This might be since CPT, given for the prevention of many opportunistic infections such as pneumocystis pneumonia, toxoplasmosis, bacterial infections & diarrheal diseases [38], might have a direct or indirect effect on retaining of patients on HIV care.

In this study, it was also found that ambulatory functional status at enrollment increases the risk of LTFU. This is in line with studies done in Ethiopia and Nigeria [35,36]. This might be due to their inability to do their routine activity including their daily work and may become socially and economically disadvantageous and this may affect their stay in care. In contrast to other studies which showed that bedridden patients are associated with lost to follow up [15,32], in this study being bed ridden patient is not associated with LTFU. This might be because of in this study there were small number of bedridden patients (very small sample size) which may result a biased statistical estimation.

Regimen type AZT-3TC-NVP during the starting of ART increases the risk of LTFU as compared to TDF-3TC-EFV. This is in line with a study done in Mekelle, Ethiopia [23]. This might be because AZT-3TC-NVP is taken twice per day (drug burden) that results in the patient not taking the drug appropriately and end up with LTFU and also TDF-3TC-EFV is fixed-dose combinations of ART agents which helps to reduce dosing complexity [45]. Tenofo-vir regimen is claimed to have a better safety profile, even though it may result renal toxicity, compared to zidovudine because of a reduced incidence of anemia and fat redistribution [46–50], so patients with AZT-3TC-NVP may have a loss of hope on the medication if they become ill due to anemia and other complications and finally may be lost from HIV care [32].

This study has some limitations. Since this study used baseline sociodemographic and clini-cal factors, there may be a change of these variables after a time but not properly recorded to include them in the analysis. This study was also conducted based on secondary data and full data on some potentially important predictors such as viral load, residence, and distance from

the hospital were not available. For the differences noted between AZT and TDF regimen, the sample size for the AZT regimen was much smaller and may have provided a statistical bias that needs to be investigated further.

## Conclusion

In this study the incidence of LTFU was high. For investigating the predictor variables of LTFU, competing risk regression analysis was done considering death as competing event since the observation of death obscure the observation of LTFU. So, in this study after considering death as a competing event, patients on ART whose age was younger, not taking CPT prophylaxis, not taking IPT prophylaxis, being daily laborer, ambulatory functional status and being taking AZT-3TC-NVP medication at the start of ART were at higher risk for LTFU. Therefore, giving more attention and close follow up of these high-risk groups could decrease the rate of LTFU.

## Supporting information

**S1 Dataset.**
(XLS)

## Acknowledgments

We are thankful for University of Gondar Comprehensive Specialized Hospital administrative bodies and chart room workers for their cooperation as well as their permission to conduct the study. We are also thankful to the physicians who work in the ART clinic and ART data managers for giving relevant information. Finally, we would like to thank data collectors and the supervisor for their tolerance and commitment.

## Author Contributions

**Conceptualization:** Achamyeleh Birhanu Teshale, Adino Tesfahun Tsegaye, Haileab Fekadu Wolde.

**Data curation:** Achamyeleh Birhanu Teshale, Adino Tesfahun Tsegaye, Haileab Fekadu Wolde.

**Formal analysis:** Achamyeleh Birhanu Teshale, Adino Tesfahun Tsegaye, Haileab Fekadu Wolde.

**Funding acquisition:** Achamyeleh Birhanu Teshale, Adino Tesfahun Tsegaye, Haileab Fekadu Wolde.

**Investigation:** Achamyeleh Birhanu Teshale, Adino Tesfahun Tsegaye, Haileab Fekadu Wolde.

**Methodology:** Achamyeleh Birhanu Teshale, Adino Tesfahun Tsegaye, Haileab Fekadu Wolde.

**Project administration:** Achamyeleh Birhanu Teshale, Adino Tesfahun Tsegaye, Haileab Fekadu Wolde.

**Resources:** Achamyeleh Birhanu Teshale, Adino Tesfahun Tsegaye, Haileab Fekadu Wolde.

**Software:** Achamyeleh Birhanu Teshale, Adino Tesfahun Tsegaye, Haileab Fekadu Wolde.

**Supervision:** Achamyeleh Birhanu Teshale, Adino Tesfahun Tsegaye, Haileab Fekadu Wolde.

**Validation:** Achamyeleh Birhanu Teshale, Adino Tesfahun Tsegaye, Haileab Fekadu Wolde.

**Visualization:** Achamyeleh Birhanu Teshale, Adino Tesfahun Tsegaye, Haileab Fekadu Wolde.

**Writing – original draft:** Achamyeleh Birhanu Teshale, Adino Tesfahun Tsegaye, Haileab Fekadu Wolde.

**Writing – review & editing:** Achamyeleh Birhanu Teshale, Adino Tesfahun Tsegaye, Haileab Fekadu Wolde.

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
