## [Decision Letter · Decision Letter 0]

25 Nov 2019

PONE-D-19-22281

Incidence and predictors of loss to follow up among adult HIV patients on antiretroviral therapy in University of Gondar Comprehensive Specialized Hospital:  A competing risk regression modeling

PLOS ONE

Dear Mr. Teshale,

Thank you for submitting your manuscript to PLOS ONE. After careful consideration, we feel that it has merit but does not fully meet PLOS ONE’s publication criteria as it currently stands. Therefore, we invite you to submit a revised version of the manuscript that addresses the points raised during the review process as shown in the two reviewer's comments.

We would appreciate receiving your revised manuscript by Jan 09 2020 11:59PM. To enhance the reproducibility of your results, we recommend that if applicable you deposit your laboratory protocols in protocols.io, where a protocol can be assigned its own identifier (DOI) such that it can be cited independently in the future. For instructions see: http://journals.plos.org/plosone/s/submission-guidelines#loc-laboratory-protocols

We look forward to receiving your revised manuscript.

Kind regards,

Matt A Price

Academic Editor

PLOS ONE

Journal Requirements:

1. We note that you have indicated that data from this study are available upon request. PLOS only allows data to be available upon request if there are legal or ethical restrictions on sharing data publicly. For information on unacceptable data access restrictions, please see http://journals.plos.org/plosone/s/data-availability#loc-unacceptable-data-access-restrictions.

2. Thank you for including your funding statement; "No"

Please provide an amended Funding Statement that declares *all* the funding or sources of support received during this specific study (whether external or internal to your organization) as detailed online in our guide for authors at http://journals.plos.org/plosone/s/submit-now.  

Please state what role the funders took in the study.  If any authors received a salary from any of your funders, please state which authors and which funder. If the funders had no role, please state: "The funders had no role in study design, data collection and analysis, decision to publish, or preparation of the manuscript."

4. Please include a caption for figure 2

Reviewers' comments:

Reviewer's Responses to Questions

**Comments to the Author**

1. Is the manuscript technically sound, and do the data support the conclusions?

Reviewer #1: Yes

2. Has the statistical analysis been performed appropriately and rigorously? 

Reviewer #1: Yes

3. Have the authors made all data underlying the findings in their manuscript fully available?

Reviewer #1: No

4. Is the manuscript presented in an intelligible fashion and written in standard English?

Reviewer #1: Yes

5. Review Comments to the Author

Reviewer #1: Please amend the statement in the conclusion to indicate that giving special attention to the high risk groups for loss to follow-up highlighted in this manuscript could decrease the rate of LTFU rather than stating it will decrease the rate of LTFU.

Please state the financial support received from the University of Gondar in the relevant financial disclosure section.

Line 60, amend to read 'Access to antiretroviral therapy has increased...'

Line 67, amend to read 'HIV infected patients...'

Line 68, correct to read 'as the ART program...'

Line 69, amend to '...is expanding; this has resulted in a decrease in the clinician-to-patient ratio.'

Line 72, please clarify if the figure 24.6% is a proportion, prevalence or incidence rate. Please also clarify if this figure is representative of the entire region or if the study was conducted in specific country or part of the region.

Line 82 should read '...constitutes...'

Line 85, does this variable refer to the level of support from the caregiver?

Line 86; please correct the spelling of baseline.

Line 110, as the guidelines internationally have changed quite a lot over the past 5 years, please outline the eligibility criteria for initiation of ART and state how this changed during the duration of the study.

Line 131 should read '...daily basis...'

Line 155; please provide a definition for the variable disclosure. Was disclosure specific or did disclosure to at least 1 individual regardless of who they were qualify?

Line 156; please specify the definition of a caregiver in this context.

Line 161; please specify how you defined 'working functional status'.

Line 262; Compared to working? Please provide the definition for 'working' and 'ambulatory' that you used in this study.

Line 162; please correct the statement to read '... were taking TDF-3TC-EFV at the start of their medication.'

Line 162; consider amending the sentence to read 'Regarding prophylaxis against opportunistic infections...'

Line 182; based on table 1, should this read '...,having an occupation other than that of a government employee,...'? Should this statement include self-employed individuals as well?

Line 182; Please correct the grammar in this paragraph. Consider 'Graphically, belonging to the age group 15-30, being single, having an occupation other than that of a government employee, not having a caregiver, not taking IPT and CPT prophylaxis, being ambulatory, not having disclosed their HIV status, a baseline WHO stage IV, being on a regimen of AZT-3TC-NVP and not having a formal education were all risk factors for LTFU'.

Line 186; this should read figure 2.

Line 222; Your study looked at data between 2015 and 2018, as you mention the possibility of the test and treat strategy having contributed to this, please clarify when the test and treat strategy was introduced in Ethiopia.

Please note that the study did not measure the contribution of workload at the clinic or the test and treat strategy on adherence to care and therefore it is out of place to state categorically in the conclusion that the test and treat strategy contributed to the higher rates of LTFU in your study.

Line 226; As the contribution of patient dissatisfaction was not measured in this study, please clarify that this will need to be measured in another study before a conclusion can be drawn on this.

Line 251; Please clarify how individual patients on IPT would note the benefits and better health outcomes from this intervention.

Are you implying that the the patients are well educated on the health benefits of IPT to the extent that it improved their adherence to ART?

Line 255; Please clarify what the national guideline on IPT and CPT was during the course of the study between 2015 and 2018. Were all patients offered the prophylactic drugs in equal measure to justify the conclusions you have drawn on motivation to remain on ART by those on prophylaxis?

Line 259; this sentence seems highly speculative and doesn't seem to be supported by the data in this paper.

Line 260; correct to read 'Holy water'.

Line 264; This statement does not explain why no association was seen between being bedridden and LTFU.

Line 270; Consider comparing the safety profile of AZT regimens vs TDF regimen overall, ie. beyond anemia.

Under limitations of the study, for the differences noted between AZT and TDF regimen, please make a note that the sample size of the former was much smaller and may have provided a statistical bias that needs to be investigated further.

Please comment in the conclusion how the crude sub-hazard ratio was affected for the various variables by the competing risk analysis identifying mortality as a competing risk to LTFU.

Line 280 has an error; patients on TDF-3TC-EFV were less likely to be lost to follow-up than those on AZT-3TC-NVP.

Please make the correction.

Correct the title of Table 1 to read HIV positive patients.

Variables differ in table 1 and table 3. Under occupation, table 1 refers to government employee, self-employee, daily labor, housewife, jobless, and other while table 3 refers to employee, daily labor and others. Please reconcile the variables in these 2 tables.

Figure 2 is not visible and neither is the attachment.

Please correct the title of figure 2 as it currently reads figure 1.

6. PLOS authors have the option to publish the peer review history of their article (what does this mean?). If published, this will include your full peer review and any attached files.

Reviewer #1: Yes: Vincent Muturi-Kioi

---

## [Author Response · Author response to Decision Letter 0]

9 Dec 2019

Editors’ comments

When submitting your revision, we need you to address these additional requirements (issues related to journal requirements) and we addressed all the issues raised that is:-

1. Please ensure that your manuscript meets PLOS ONE's style requirements, including those for file naming. So, based on the link what you have given we adjust our manuscript based on PLOS ONE's style. 

2. Regarding to questions raised about data availability and funding statement we amended and included in the revised cover letter.

3. PLOS requires an ORCID iD for the corresponding author in Editorial Manager on papers submitted after December 6th, 2016, so the author created ORCID iD

4. Please include a caption for figure 2, we checked and corrected (we added the caption) in the revised work.

Reviewers’ comments and authors response 

Dear reviewer, we would like to say thank you for the constructive comments given for the betterment of our paper. Here are the comments raised and the authors response. 

1. Please amend the statement in the conclusion to indicate that giving special attention to the high-risk groups for loss to follow-up highlighted in this manuscript could decrease the rate of LTFU rather than stating it will decrease the rate of LTFU. 

Authors response; checked and Amended from …giving special attention and close follow up of these high-risk groups will decrease the rate of LTFU…to …giving special attention to the high-risk groups for lost to follow up highlighted in this study could decrease the rate of LTFU.

2. Please state the financial support received from the University of Gondar in the relevant financial disclosure section. 

Authors response; the funding statement is also commented above as a journal requirement and we include in the cover letter.

3. Line 60, amend to read 'Access to antiretroviral therapy has increased...' 

Authors response; checked and amended from ‘...Antiretroviral therapy has increased rapidly…’ to… ‘Access to antiretroviral therapy has increased rapidly…’ in the revised manuscript 

4. Line 67, amend to read 'HIV infected patients...'

Authors response; checked and amended from ‘…. Human Immune Deficiency Virus (HIV) infected patients…’ to '…. HIV infected patients...’

5. Line 68, correct to read 'as the ART program...' 

Authors response; checked and corrected from ‘ART program’ to ‘the ART program’ the revised manuscript

6. Line 69, amend to '...is expanding; this has resulted in a decrease in the clinician-to-patient ratio.' 

Authors response; Checked and Amended or modified from…. ‘is expanding that results decreasing of clinician-to-patient ratio’ to … ‘is expanding; this has resulted in a decrease in the clinician-to-patient ratio’.

7. Line 72, please clarify if the figure 24.6% is a proportion, prevalence or incidence rate. Please also clarify if this figure is representative of the entire region or if the study was conducted in specific country or part of the region. 

Authors response; Here the figure 24.6% is a proportion and this is representative of the specific country Uganda which is one of the countries in sub Saharan Africa. This is modified in the revised manuscript as …’According to a study done in one of the countries in Sub Saharan Africa, Uganda the proportion of LTFU is 24.6 %’. 

8. Line 82 should read '...constitutes...' 

Authors response; Checked and amended from ‘…constitute to ‘. ...constitutes...’ in the revised manuscript.

9. Line 85, does this variable refer to the level of support from the caregiver? 

Authors response; it means whether a patient has a care giver or not (yes/no), without considering the level of support, which is recorded as like this in ART intake form.

10. Line 86; please correct the spelling of baseline. 

Authors response; initially it was ‘base line’ but now in revised manuscript we correct the spelling into ‘baseline’

11. Line 110, as the guidelines internationally have changed quite a lot over the past 5 years, please outline the eligibility criteria for initiation of ART and state how this changed during the duration of the study. 

Authors response; Even though the ART guide line changed quite a lot in the past five years, regarding the eligibility criteria for initiation of ART (now which is test and treat) as well. But in line 110 ‘…. from the ART database according to their entry time to the follow-up, age and eligibility criteria’, when we say the eligibility criteria it is to mean inclusion criteria (to include those patients who had at least one follow up) and now it is amended for clarity in the revised manuscript as ‘….from the ART database according to their entry time to the follow-up, age and inclusion criteria’.

12. Line 131 should read '...daily basis...'

Authors response; amended from ‘...daily base...’ before to ‘...daily basis...’ in the revised manuscript.

13. Line 155; please provide a definition for the variable disclosure. Was disclosure specific or did disclosure to at least 1 individual regardless of who they were qualify? 

Authors response; Disclosure in this study was to mean disclosure of the status that is being HIV positive to at least one individual regardless of who they were qualify (asked like …. ‘Does anyone else know about your HIV status?’ and recorded in the ART intake form). For clarity we added it in the method section in the revised manuscript.

14. Line 156; please specify the definition of a caregiver in this context. 

Authors response; in this study caregiver was defined as anyone who can support or give care to the individual with HIV and it was recorded as yes/no in the ART intake form. For clarity it is also stated in the method section in the revised manuscript. 

15. Line 161; please specify how you defined 'working functional status'.

Line 262; Compared to working? Please provide the definition for 'working' and 'ambulatory' that you used in this study.

Authors response; The functional status was defined based on the ART guide line; Working: able to perform usual work inside or outside home, Ambulatory: able to perform activity of daily living. Bedridden: not able to perform activity of daily living and now in the revised manuscript it is stated in the method section.

16. Line 162; please correct the statement to read '... were taking TDF-3TC-EFV at the start of their medication.' 

Authors response; We amended it …’were taken TDF-3TC-EFV at the start of their medication’ initially to ‘…were taking TDF-3TC-EFV at the start of their medication.' In the revised manuscript. 

17. Line 162; consider amending the sentence to read 'Regarding prophylaxis against opportunistic infections...' 

18. Authors response; we checked the sentence and we modified ‘…. Regarding opportunistic prophylaxis’ to 'Regarding prophylaxis against opportunistic infections...' in the revised manuscript.

19. Line 182; based on table 1, should this read '..., having an occupation other than that of a government employee,'? Should this statement include self-employed individuals as well? Authors response; Yes, this includes both government and self-employed patients. That is both governments employed and self-employed … counted as employed

20. Line 182; Please correct the grammar in this paragraph. Consider 'Graphically, belonging to the age group 15-30, being single, having an occupation other than that of a government employee, not having a caregiver, not taking IPT and CPT prophylaxis, being ambulatory, not having disclosed their HIV status, a baseline WHO stage IV, being on a regimen of AZT-3TC-NVP and not having a formal education were all risk factors for LTFU'. Authors response; we reread it and we modified the original paragraph stating ‘Graphically, being age group 15-30, being single, being occupation other than employee, being not having caregiver, not taking IPT and CPT prophylaxis, being ambulatory, those who have not disclosed their HIV status, being baseline WHO stage IV, being taking regiment type AZT-3TC-NVP and those who have no education were at higher risk of LTFU)’ to ‘Graphically, being age group 15-30, being single, being occupation other than employee, being not having caregiver, not taking IPT and CPT prophylaxis, being ambulatory, those who have not disclosed their HIV status, being baseline WHO stage IV, being taking regiment type AZT-3TC-NVP and those who have no education were at higher risk of LTFU’ in the revised manuscript. 

21. Line 186; this should read figure 2. 

Authors response; Checked and modified from saying figure 1 to figure 2 in the revised manuscript.

22. Line 222; Your study looked at data between 2015 and 2018, as you mention the possibility of the test and treat strategy having contributed to this, please clarify when the test and treat strategy was introduced in Ethiopia. 

Please note that the study did not measure the contribution of workload at the clinic or the test and treat strategy on adherence to care and therefore it is out of place to state categorically in the conclusion that the test and treat strategy contributed to the higher rates of LTFU in your study.

23. Line 226; As the contribution of patient dissatisfaction was not measured in this study, please clarify that this will need to be measured in another study before a conclusion can be drawn on this.

Authors response; Based on the comments and issues raised for line 222 and 226 we write the justification accordingly like this ‘…. Lost to follow up in our study turned out to be higher than the study done in Mekelle might be due to the study period in which currently, starting from 2016, test and treat strategy is introduced in Ethiopia and this might increase the number of patients on follow up and which in turn increases self-referral to the other health facilities, as investigated by different studies. In addition, in search of literatures, patient satisfaction with healthcare system is associated with engagement in HIV care, so there may be the difficulty of appropriate health service delivery and access since this hospital is a Comprehensive Specialized Hospital, which serves many patients and finally results in patient dissatisfaction and loss to follow up. Furthermore, the way operationalizing LTFU might be a reason for higher LTFU in our study as compared to studies in India and Asia because in our study LTFU is defined when a patient is lost for at least 90 days but in studies in India and Asia LTFU is defined when the patient is lost for at least 180 days.’

24. Line 251; Please clarify how individual patients on IPT would note the benefits and better health outcomes from this intervention.

Are you implying that the patients are well educated on the health benefits of IPT to the extent that it improved their adherence to ART?

Authors response; it is not to mean patients are well educated on the health benefits of IPT to the extent that it improved their adherence to ART but it is to mean taking IPT based on the recommendations of the national ART guideline prevent the most devastating and common illness among HIV patients which is Tuberculosis, so preventing tuberculosis improves the health as well makes the patient well. 

25. Line 255; Please clarify what the national guideline on IPT and CPT was during the course of the study between 2015 and 2018. Were all patients offered the prophylactic drugs in equal measure to justify the conclusions you have drawn on motivation to remain on ART by those on prophylaxis?

Authors response; all patients do not offered the prophylactic drugs, it depends on the patients profile that is IPT is administered at enrolment to HIV care after ruling out active TB but CPT is given for all clients any WHO stage and CD4 count <=350 cells/mm3 Or WHO 3 or 4 irrespective of CD4 level. But many individuals expect burden of medication as a direct or indirect cause for discontinuing the medications and LTFU. So, our intention was to investigate this issue. In the revised manuscript we modify and re write the justification by considering the issues raised. 

26. Line 259; this sentence seems highly speculative and doesn't seem to be supported by the data in this paper and Line 260; correct to read 'Holy water'.

Authors response; we amended the paragraph by removing unnecessary justifications and in the revised manuscript we removed the paragraph ‘..But if patients are not taking CPT, they are more vulnerable to many opportunistic infections and finally, they may be catchup with such diseases and either they may develop drug toxicity due to drug-drug interaction or they may prefer to go to Holly water by discontinuing such burden of drugs and finally end up with LTFU’ and we re write the justification.

27. Line 264; This statement does not explain why no association was seen between being bedridden and LTFU.

Authors response; Now it is explained in the main manuscript as ‘In contrast to other studies which showed that bedridden patients are associated with lost to follow up, in this study being bed ridden patient is not associated with LTFU. This might be because of in this study there were small number of bedridden patients (very small sample size) which may result a biased statistical estimation. 

28. Line 270; Consider comparing the safety profile of AZT regimens vs TDF regimen overall, ie. beyond anemia.

Authors response; we try to adjust it to include the safety profile beyond anemia by searching different literatures. In the revised manuscript we rewrite it as ‘…..Tenofovir regimen is claimed to have a better safety profile, even though it may result renal toxicity, compared to zidovudine because of a reduced incidence of anemia and fat redistribution, so patients with AZT-3TC-NVP may have a loss of hope on the medication if they become ill due to anemia and other complications and finally may be lost from HIV care.’ 

29. Under limitations of the study, for the differences noted between AZT and TDF regimen, please make a note that the sample size of the former was much smaller and may have provided a statistical bias that needs to be investigated further.

Authors response; As limitation, we indicated for the differences noted between AZT and TDF regimen, as ‘…the sample size for the AZT regimen was much smaller and may have provided a statistical bias that needs to be investigated further.’

30. Please comment in the conclusion how the crude sub-hazard ratio was affected for the various variables by the competing risk analysis identifying mortality as a competing risk to LTFU.

Authors response; Based on the issues raised here we modified our conclusion as ‘in this study the incidence of LTFU was high. For investigating the predictor variables of LTFU, competing risk regression analysis was done considering death as competing event since the observation of death obscure the observation of LTFU. So, in this study after considering death as a competing event, patients on ART whose age was younger, not taking CPT prophylaxis, not taking IPT prophylaxis, being daily laborer, ambulatory functional status and being taking AZT-3TC-NVP medication at the start of ART were at higher risk for LTFU. Therefore, giving more attention and close follow up of these high-risk groups could decrease the rate of LTFU.’

31. Line 280 has an error; patients on TDF-3TC-EFV were less likely to be lost to follow-up than those on AZT-3TC-NVP. Please make the correction.

Authors response; we checked and we amended it as ‘…. those on AZT-3TC-NVP were at higher risk of LTFU.’

32. Correct the title of Table 1 to read HIV positive patients. 

Authors response; Checked and corrected in the revised manuscript

33. Variables differ in table 1 and table 3. Under occupation, table 1 refers to government employee, self-employee, daily labor, housewife, jobless, and other while table 3 refers to employee, daily labor and others. Please reconcile the variables in these 2 tables. 

Authors response; We were recategorized the variable occupation because of not fulfilling the chi square assumption for analysis but now we reconcile it by changing it in the descriptive part (Table1) in the revised manuscript.

34. Figure 2 is not visible and neither is the attachment. 

Authors response; Checked and corrected in the revised manuscript

35. Please correct the title of figure 2 as it currently reads figure 1. 

Authors response; Checked and corrected and now it reads figure 2

---

## [Editor Report · Decision Letter 1]

20 Dec 2019

Incidence and predictors of loss to follow up among adult HIV patients on antiretroviral therapy in University of Gondar Comprehensive Specialized Hospital:  A competing risk regression modeling

PONE-D-19-22281R1

Dear Dr. Teshale,

We are pleased to inform you that your manuscript has been judged scientifically suitable for publication and will be formally accepted for publication once it complies with all outstanding technical requirements.

With kind regards,

Matt A Price

Academic Editor

PLOS ONE
---

## [Editor Report · Acceptance letter]

8 Jan 2020

PONE-D-19-22281R1 

Incidence and predictors of loss to follow up among adult HIV patients on antiretroviral therapy in University of Gondar Comprehensive Specialized Hospital:  A competing risk regression modeling 

Dear Dr. Teshale:

I am pleased to inform you that your manuscript has been deemed suitable for publication in PLOS ONE. Congratulations! Your manuscript is now with our production department. 

With kind regards,

on behalf of

Dr. Matt A Price 

Academic Editor

PLOS ONE